# Long-term wage inequality in imperial China: From 202 BCE to 1912 CE

**Qiang Wu[1], Guangyu Tong[2]\*, Peng Zhou[3]\***

**1** University of International Business and Economics, Beijing, China, **2** Yale School of Medicine, Yale University, New Haven, Connecticut, United States of America, **3** Cardiff Business School, Cardiff University, Cardiff, United Kingdom

\* zhoup1@cardiff.ac.uk (PZ); guangyu.tong@yale.edu (GT)

**Data Availability Statement:** All relevant data are within the paper and its Supporting Information files.

**Funding:** This study was supported by the General Project of the National Social Science Fund of China (Grant Number: 23BJY133). The funders had no role in study design, data collection and

## Abstract

This paper attempts to describe and explain the long-term evolution of wage inequality in imperial China, covering over two millennia from the *Han* dynasty to the *Qing* dynasty (202 BCE-1912 CE). Based on historical government records of official salaries, commodity prices, and agricultural productivity, we convert various forms of salaries to equivalent rice volumes and comparable salary benchmarks. Wage inequality is measured by salary ratios and (partial) Gini coefficients between official and peasant classes as well as within the official class. The inter-class wage inequality features an "inverted U-u" pattern—first rose before the *Tang* dynasty and then declined afterwards (the "inverted U" trends) with "inverted u" dynastic cycles. The intra-class wage inequality has a secular decline trend. We propose a unified framework incorporating technological, institutional, political, and social (TIPS) mechanisms to explain both long-term and short-term patterns. It is concluded that the technological mechanism dominated the rise of wage inequality, while the political mechanism (emperor-bureaucracy power tensions) drove the decline.

## Introduction

The growth-inequality nexus is a prominent research interest among economists. A pioneering yet controversial view is the Kuznets hypothesis which proposes that inequality follows an inverted U-shaped curve as the economy transitions from an agrarian to an industrial society in modern growth [1]. Later studies extend the hypothesis to the premodern growth as a "super Kuznets curve" [2]. Some studies show support for such an inverted U relationship under certain conditions (e.g., [3,4]), while many others refute a clear relationship between productivity and inequality (e.g., [5–8]). As a salvage attempt, Milanovic argues that there may not be a single Kuznets curve for a given country but rather a series of Kuznets curves, or waves, responding to new technological revolutions [9]. This is known as the Kuznetsian view. However, the prevailing tendency in literature is to discard the Kuznets hypothesis as obsolete [10]. As Kuznets himself commented, the hypothesis is "*perhaps 5 percent empirical information and 95 percent speculation, some of it possibly tainted by wishful thinking*".

analysis, decision to publish, or preparation of the
manuscript.

**Competing interests:** The authors have declared
that no competing interests exist.

The debate has nonetheless inspired numerous theories of the growth-inequality relationship such as innovation and technological changes [11,12], industrial structures [13,14], employment structures [15,16], characteristics of labor markets [17,18], urbanization [19], and globalization [20]. Specifically, Piketty and his coauthors challenged the reliance on technology in explaining inequality changes [21–23]. Based on evidence from France, Australia, Canada, Britain, and the US in the 20[th] century, they find that social norms (implicit rules) and institutional arrangements (explicit rules) in distribution (e.g., union) and redistribution (e.g., taxation) can play more important roles in shaping inequality [24–26]. Nevertheless, most studies are based on postwar evidence, and few have been devoted to long-term interdependence between growth and inequality over several centuries [25].

In contrast to the burgeoning literature on Unified Growth Theory [26,27], the advancement in "Unified Inequality Theory" is slow. One reason is the lack of reliable data on historical inequality. In contemporary society, income survey data can meet the demand to construct a comprehensive Gini coefficient. For pre-industrial society, social tables are sometimes used, but within-class inequalities cannot be well captured [28]. As a result, research is usually done for subperiods of history [29] and mainly in the west, e.g., Germany [30], France [31], Russia [32], Britain [33], and the US [34]. Dissimilar to Western countries, China has no continuously marketed wage data, just fragmentary records of official salaries [35] and construction laborers [36], so the research on inequality in China mainly concentrates on a particular dynasty [37]. To broaden the geographical and temporal scopes, this study compiles various historical records and provides useful benchmarks of wage inequality in imperial China. It covers an extensive period of 2,113 years, spanning from 202 BCE (the beginning of the *Han* dynasty) to 1912 CE (the end of the *Qing* dynasty).

China offers an interesting case for studying economic history due to its cultural continuity and geographical isolation [38]. A wealth of historical records on wages and prices has been meticulously documented as the Chinese civilization evolved over the past several millennia. Despite some changes in the Chinese language, the written form has remained largely stable, allowing any educated individuals in the 21[st] century to comprehend records written more than two thousand years ago. This linguistic continuity enables us to construct a long series of measures of wage inequality, a task that proves much more challenging in other countries. To facilitate exposition, Fig 1 lists all Chinese dynasties chronologically with historical events in the western civilization. It can be a helpful navigator for those who are not familiar with Chinese dynasties.

The empirical benchmarks on wage inequality are the first contribution of our paper. Based on historical records on official salaries, prices, and agricultural productivity, we measure wage inequality both between the rich (officials) and the poor (peasants) as well as among the rich. Admittedly, land rents and corrupt revenue lead to underestimation of official income, making our measure of **wage inequality** a lower bound of **income inequality**. In addition, our data cannot capture all aspects of inequality in imperial China due to the lack of records in every social class (e.g., seigniors, artisans, merchants, soldiers) and every region, so the wage ratios and Gini coefficients computed in this paper are a *partial* rather than a comprehensive measure of inequality. We argue that officials and peasants represent the majority of the rich and the poor, so the constructed measures based on the two social classes ("inter-class" inequality) allow us to portray, at the very least, a rough picture of inequality in imperial China. Overall, we find a gradual increase in wage inequality from the *Han* to the mid-*Tang* dynasty (about a millennium from 202 BCE-841 CE) and a decline from the late-*Tang* to the *Qing* dynasty (about a millennium from 841–1912 CE). The trends appear to be in line with the inverted U shape pattern. In addition to the long-term pattern, we also find a short-term pattern of "inverted u" dynastic cycles (the lowercase "u" is intentionally used for cycles and

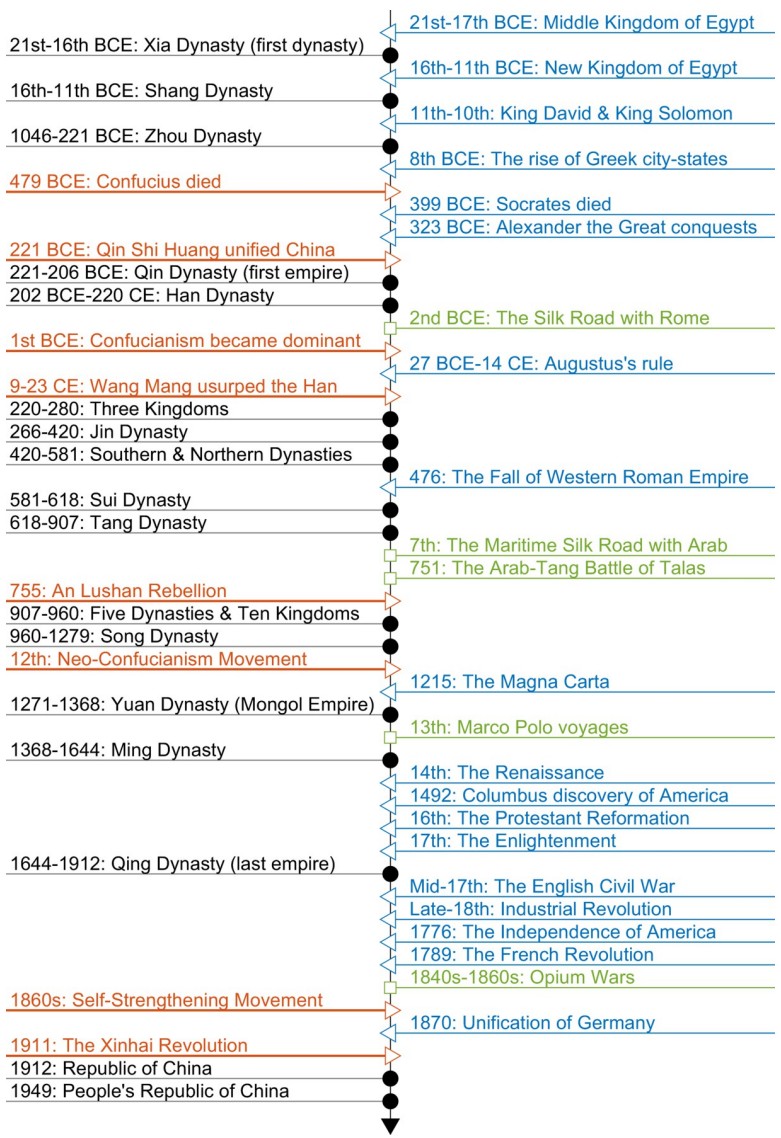

21st-16th BCE: Xia Dynasty (first dynasty)

21st-17th BCE: Middle Kingdom of Egypt

16th-11th BCE: Shang Dynasty

16th-11th BCE: New Kingdom of Egypt

1046-221 BCE: Zhou Dynasty

11th-10th: King David & King Solomon

479 BCE: Confucius died

8th BCE: The rise of Greek city-states

399 BCE: Socrates died
323 BCE: Alexander the Great conquests

221 BCE: Qin Shi Huang unified China
221-206 BCE: Qin Dynasty (first empire)
202 BCE-220 CE: Han Dynasty

2nd BCE: The Silk Road with Rome

1st BCE: Confucianism became dominant

27 BCE-14 CE: Augustus's rule

9-23 CE: Wang Mang usurped the Han
220-280: Three Kingdoms
266-420: Jin Dynasty
420-581: Southern & Northern Dynasties

476: The Fall of Western Roman Empire

581-618: Sui Dynasty
618-907: Tang Dynasty

7th: The Maritime Silk Road with Arab
751: The Arab-Tang Battle of Talas

755: An Lushan Rebellion
907-960: Five Dynasties & Ten Kingdoms
960-1279: Song Dynasty
12th: Neo-Confucianism Movement

1215: The Magna Carta

1271-1368: Yuan Dynasty (Mongol Empire)

13th: Marco Polo voyages

1368-1644: Ming Dynasty

14th: The Renaissance
1492: Columbus discovery of America
16th: The Protestant Reformation
17th: The Enlightenment

1644-1912: Qing Dynasty (last empire)

Mid-17th: The English Civil War
Late-18th: Industrial Revolution
1776: The Independence of America
1789: The French Revolution
1840s-1860s: Opium Wars

1860s: Self-Strengthening Movement

1870: Unification of Germany

1911: The Xinhai Revolution
1912: Republic of China
1949: People's Republic of China

**Fig 1.**

the uppercase "U" is used for trends). Therefore, the overall wage inequality features an inverted U-u pattern. By contrast, the wage inequality within officials ("intra-class" inequality) witnessed a secular decline over the two millennia, but with inverted u cycles.

The second contribution of our paper is theoretical. Building on the long-term series of (partial) wage inequality, we propose a unified framework to explain the evolution of inequality in a pre-industrial society. Our explanation combines the Kuznetsian productivity view with social, institutional, and political views. The initial increase in inequality was driven by technological improvements in agriculture [39], while the later decline was mainly due to political tensions. Changes in explicit rules (e.g., institutionalization of imperial exams) and implicit rules (e.g., Neo-Confucianism social norms) were embodiments of the emperor-bureaucracy tension. Specifically, Neo-Confucianism in the *Song* dynasties (from circa 12[th] century) introduced new moral principles discouraging individual economic pursuits while emphasizing the supreme authority of emperors. These new social norms were supported by

emperors to strengthen their power and had permanent impacts to wage inequality [40]. Moreover, bottom-up uprisings, internal rebellions, and external wars resulted in lower wage inequality in later dynasties and in later phases of each dynasty because the military budget was prioritized over official salaries. The novelty of our paper is that long-term "inverted U trends" and short-term "inverted u cycles" can be explained using the same framework, which is even generalizable to the modern world.

This paper is organized as follows. Section 2 describes the social and income hierarchies in imperial China. In Section 3, we describe our data and measures of wage inequality. Section 4 presents the results of wage inequality and identifies patterns of wage inequality. In Section 5, we propose a unified theoretical framework to explain the identified patterns and Section 6 concludes.

## Social hierarchy and income ladder

Scheidel argues for a "Great Convergence" between the West and China that spanned the entire first millennium BCE and the first half of the first millennium CE, until a "Great Divergence" began to unfold from about the 6[th] century CE onward [41]. The Western civilization is characterized by a tradition of *decentralized* governance [42]. This tradition was inherited from ancient Greek city-states and developed under the Roman Empire (e.g., a balance of power among the Senate, magistrates, and provinces), the Middle Ages (e.g., shared power among central monarchs, feudal lords, and local nobility), the Renaissance and Enlightenment periods (e.g., philosophical ideas of individual rights, freedoms, and the social contract), and modern times (e.g., federal systems in the US, devolved governments in the UK, and the Subsidiarity Principle in the EU). In contrast, *centralized* governance is a prominent characteristic of eastern civilizations [43], notably exemplified by China [44] and Japan [45]. An intended outcome of the high concentration of power is a stable social hierarchy paired with a stable income distribution system founded in the *Qin* and *Han* dynasties [46].

The social hierarchy in ancient China has a long history originated in the *Zhou* dynasty (1046–221 BCE). It is represented by a top-down pyramid of power (Fig 2) with an isomorphic structure between family and government.

At the top of the pyramid is the *emperor* (the "Son of Heaven", the supreme ruler, and the embodiment of authority), whose legitimacy is believed to be granted by Heaven. Directly below the emperor were aristocrats (nobles with hereditary status) and seigniors (sons of former emperors without legitimate claims to the throne). In early dynasties (e.g., *Zhou*, *Han*, *Jin*), they served as high-ranking officials, military commanders, and advisers to the emperor. In later dynasties, however, most of seigniors did not serve as officials due to increasing political tensions among imperial princes. For example, in the *Ming* dynasty, the number of seigniors grew very quickly because there was no limit on their fertility. Central and local governments financially supported them, so the fiscal burden also became very large when their population grew. We do not have accurate data for this social class, but their wage can be reasonably assumed to be close to the low-ranking officials.

Further down the hierarchy were the *scholars*, known as the *Shi* class, who were either sons of the aristocrats without hereditary status or talents from the lower class. They connected the upper and the lower classes, so they are in some sense the "middle class" in imperial China. Before the *Sui* dynasty, most scholars obtained their government positions through recommendations from aristocrats and other officials ("*Ju Xiao Lian*"). During the *Sui* dynasty, a new system of imperial civil service examinations ("*Ke Ju*") was introduced, becoming the primary means of selecting officials from scholars and lower class. Once they obtained an official position, the rank of the position determined their authority, responsibility, and income.

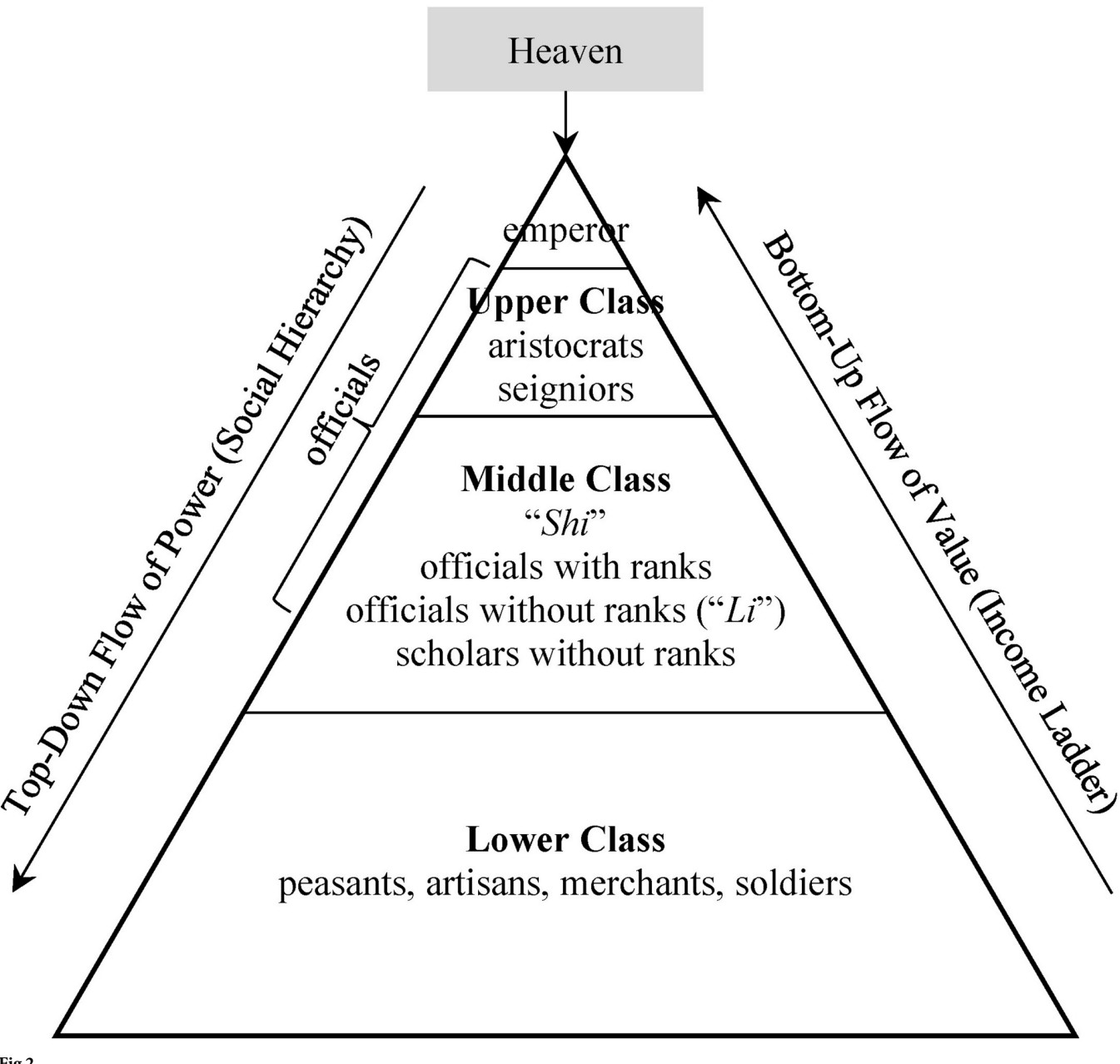

**Fig 2.**

Besides, many grassroots officials did not have ranks (called "*Li*"). They were clerks responsible for maintaining records, handling paperwork, managing finances, and performing other administrative duties. In addition to serving within the bureaucratic system, scholars in ancient China had the option to pursue purely academic endeavors. They often established and managed Confucius academies where they taught younger scholars. Leaders of Neo-Confucianism, like Zhu Xi and Lu Jiuyuan, were prominent examples of such scholars. Although they were not government officials themselves, their schools of thought held significant political influence since many officials were their students.

At the bottom of the pyramid were peasants, artisans, merchants, and soldiers. In all households of emperors, aristocrats, scholars, and ordinary people, the eldest legitimate son usually inherited the title from his father, while other sons left the households with lower or no titles. The top-down power structure of the patriarchal hierarchy effectively determines the income ladder as the centralized system decides "who gets what". The output, produced by the lower class but distributed by the upper class, forms a bottom-up flow of value. A higher social class on the income ladder receives higher income.

It is worth emphasizing that the social classes in early dynasties (prior to *Sui*) were largely frozen and usually could not be changed. For example, in the *Qin* and *Han* dynasties, a peasant was always a peasant, and his son was also a peasant. They could not freely choose to be a scholar, artisan, merchant, or soldier, and vice versa. Government officials were mainly selected from scholars and aristocrats. In other words, there was limited horizontal social mobility across occupations. Moreover, vertical social mobility across hierarchies was also restricted. For scholars (the *Shi* class), they must engage in fierce competition to ascend to higher ranks, occasionally to aristocratic titles. Only in exceptional cases can people from the lower class be promoted, e.g., a soldier with remarkable military achievements can be awarded an aristocratic title. Entrepreneurship was generally considered despicable in imperial China [47]. Most dynasties cherished the physiocratic model of economy that emphasized the importance of agriculture and restricted commercial activities. Social mobility in later dynasties (*Sui*, *Tang*, *Song*, *Ming*, and *Qing*) was significantly improved thanks to the "*Ke Ju*" system [48]. For example, if one had passed the imperial examination at the provincial level, he could become a middle-rank official. While the "*Ke Ju*" system improved intergenerational mobility, the evolution of social norms beginning in the *Song* dynasty may have restricted intragenerational mobility, as the rights of individuals were curtailed. This trend aligns with the findings of Yang & Zhou, who suggest that increasing inequality in this period was positively correlated with reduced intragenerational mobility (the Great Gatsby curve) [49].

According to historical fiscal records, officials constituted less than 1% of the total population in all dynasties. Officials received secured salaries from governments, much higher than peasants. They were the top earners in imperial China and their wage differentials significantly influence the wage inequality.

The official ranks had a steep hierarchical structure that resulted in substantial wage inequality among officials. Before the *Sui* dynasty, the ranking system distinguished 17–20 levels of officials. The highest-rank officials were paid 10,000 *dan* grains per year (hence the official rank was referred to as the *Ten Thousand Dan*), while the lowest-rank officials were only paid 100 *dan* grains per year (hence the official rank was referred to as *One Hundred Dan*). The term "*dan*", also called picul or tam, was an ancient unit of weight (equal to 29.5–31 kilograms in the *Han* dynasty and 50 kilograms after the *Song* dynasty). From the *Sui* dynasty onward, the ranking system was reformed; officials were ranked from the lowest 9th *Pin* to the highest 1st *Pin*. The term "*Pin*" means rank in Chinese. The switch from the absolute rank (*Dan*) to the relative rank (*Pin*) can better accommodate productivity changes and make salary adjustments.

The two ranking systems are comparable. For example, county governors were usually ranked at the *Six Hundred Dan* before the *Sui* dynasty but were ranked at the 7th *Pin* in later dynasties. The lowest-rank 9th *Pin* was equivalent to the *One Hundred Dan*, and the highest-rank 1st *Pin* was equivalent to the *Ten Thousand Dan*. These ranks may have different terms, but the corresponding positions can be mapped.

Official salaries took various forms including grain, salt, fabrics, labor, land (rents), copper, silver, and paper money. Earlier dynasties usually paid officials with grains and coins, whereas later dynasties increasingly used monetary payment. During periods of warfare, governments

often offered more diverse forms of payment, such as in the *Southern & Northern* dynasties when official salaries mainly took the form of fabrics, laborers, and lands (see Table 1). It is also worth emphasizing that these records were officially documented salaries which did not include hidden or corrupt revenues. Previous studies have shown that corruption in the *Ming* and *Qing* dynasties was more severe than early dynasties [50,51], so the measures based on these records are lower bounds of wage inequality.

## Empirical methods

This section starts with a discussion of data sources of records and inclusion criteria of officials. Our principle prioritizes validity over coverage. Based on these datasets, we develop two types of quantitative measures of inter-class and intra-class wage inequality.

## Data

Many studies have attempted to estimate official salaries of different dynasties in real terms [55,64–66]. In general, these studies have shown that the *Tang* dynasty had the highest official salaries, whereas the *Ming* and *Qing* dynasties had lower official salaries [56,57].

However, discrepancies do exist due to heterogeneous data sources of records and inclusion criteria of officials. For example, Wang claims that the *Han* dynasty offered relatively low official salaries based on some secondary data of grassroots officials [66]. In contrast, Zhang studies mid-to-high-ranking officials in the early *Han* dynasty and claims that *Han* officials had relatively high salaries compared to other dynasties [67]. Wang includes grassroots officials in the calculation of official wages and observes significant gaps in salaries between different levels of officials [66]. Zhu estimates salaries of officials at all levels in the *Jin* and *Southern & Northern* dynasties based on folk stories and poems [53]. His results are inconsistent with Jing who extracts data from more reliable historical archives [68]. Studies using observations at different time points of the *Song* dynasty do not reach a consensus on the true level of salaries. Gong shows that official salaries in the *Song* dynasty were among the highest in the Chinese history [65], whereas Mu argues that the average official salaries in the *Song* dynasty was only one fifth of that in the *Tang* dynasty [69]. Zhang divides the *Song* dynasty into four periods

**Table 1. Forms of official salary and data sources.**

| Dynasty | Years | Forms | Sources | References |
|---|---|---|---|---|
| Han | 202 BCE | money | Vol. 19, Book of Han | [52] |
| Han | 50 CE | grain, money | Vol. 1, Book of the Later Han | [52] |
| S & N | 550 | fabrics, labor | Vol. 27, Book of Sui | [53] |
| Sui | 581 | grain, labor, land | Vol. 24, 28, Book of Sui | [54] |
| Tang | 627, 666, 736 | grain | Vol. 90–92, Tang Hui Yao; Vol. 19, 35, Tong Dian; Vol. 55, New Tang History | [55] |
| Tang | 773, 788, 841 | land, money | Vol. 3, Six Dictionaries of Tang; Vol. 505–506, Ce Fu Yuan Gu Tianbao Lingshi, "Table of Tang" | [56] |
| Song | 960, 1056, 1080 | gran, land, money, labor, silk, salt | Vol. 163–172, History of Song | [57–59] |
| Yuan | 1285 | money | Vol. 96, History of Yuan | [60] |
| Yuan | 1320 | grain, money | Vol. 2, Yuan Mi Shu Lan Zhi | [61] |
| Ming | 1371, 1387 | grain | Vol. 60, 130,185, Records of Hongwu; Vol. 39, Ming Hui Dian | [62] |
| Ming | 1380, 1552, 1573, 1628 | grain, money | Vol. 4, 7, Duchayuan Records of Nanjing; Vol. 7, Taichang Xu Kao; Vol. 8, Siyiguan Zengding Guanze | [63] |
| Qing | 1644 | silver | Vol. 249, Huidian Cases of Qing; Vol. 71, Records of Qing Shizu | [61] |
| Qing | 1653, 1736, 1906 | grain, silver | Vol. 42, Wenxian Tongkao; Vol. 1, Jiang Chun Lin Ji | [64] |

and observes an increasing trend of salaries [57]. It is found that in most periods of the *Song* dynasty officials received at least as high salaries as those in the *Tang* dynasty. Studies on the *Ming* and *Qing* dynasties have fewer inconsistencies. Nevertheless, Hu finds that the average salary of the *Ming* dynasty is 11.75 times higher than that of ordinary people, which was not low compared to other dynasties [63]. Wang also notices the existence of a significant amount of unreported income and argues that the actual income of *Ming* officials might be underestimated [62].

To address this debate, we must ensure validity and comparability of data sources. For validity, we only use historical records and documents kept by central governments and exclude literary works, a single specific story, and local records. For comparability, we restrict the sample from the *Han* dynasty to the *Qing* dynasty (202 BCE-1912 CE). During the 2,113 years, the hierarchical structure of government system in ancient China had been relatively stable, which determined the authority, responsibilities, and incomes of each rank of officials. Before the *Han* dynasty, the *Shi* class followed a very different wage system. For instance, during the early *Zhou* dynasty (1046–475 BCE), most officials held a noble title, so their income was mainly from rents of their inherited lands. In the late *Zhou* dynasty (475–221 BCE), the official system transited from the hereditary system to professional bureaucracy, but there were many states with myriads of co-existing salary systems. Although the *Qin* dynasty (221– 206 BCE) was the first empire with a unified official ranking system, it only lasted 15 years. There were no historical records available for analysis. Thus, our sample starts with the *Han* dynasty, from when the rankings systems were generally comparable, and more data became available. The ending year 1912 marks the fall of imperial China and the beginning of the republic era (see Fig 1 for the timeline).

In terms of the subjects covered in this study, we exclude the emperor's income/wage due to the lack of available data. As the wealthiest individual in the empire, their exclusion makes our estimated wage inequality a conservative lower bound for imperial China. We also exclude aristocrats without official ranks, as most aristocrats were high-ranking officials and are already included in the data. Additionally, women's income/wage is omitted from the analysis. In a patriarchal society, women held a subservient position to men, and they generally did not have independent revenue if their brothers, husbands, or sons within the household were alive. This gender hierarchy was reinforced by Neo-Confucianism [40]. Following these principles and criteria, we have collected 25 salary tables spanning over 2000 years from reliable historical records, academic articles, and books (listed in Table 1).

## Measures

We use equivalent volumes of rice to measure the real wage. As an agricultural society, rice was the primary commodity in imperial China, so it is an ideal commensurable numeraire. Other types of payments, such as silver, copper coins, and paper money, can be subject to price fluctuations, inflations, and silver import [70,71]. However, it is important to note that rice yields would have increased over time due to agricultural advancements, which could potentially affect the measurement of both peasants' income and officials' land rent income. To address this, it is essential to clarify that agricultural output was typically distributed between officials and farmers through land rent. As unit land yields increased, both groups would have been affected proportionally since land rent captured a fixed share of the total agricultural output.

The price data are retrieved from the same sources to keep consistency when translating different forms of salary into equivalent volumes of rice. For instance, the main agriculture product, millet, in the *Han* dynasty could be converted to rice according to the exchange ratio of 10:6, which means the value of 10 *dan* millet was equivalent to 6 *dan* of rice [56,65]. Also,

the income from official land was calculated as 35 *dan* per hectare in the *Tang* and *Song* dynasties. According to Zhang, the yield of farmland per hectare was 100 *dan* [57]. Assume that the rental rate *r* = 50% of production, and one *dan* grain was equivalent to 0.7 *dan* rice. The rent of one hectare was thus 100 × 0.5 × 0.7 = 35 *dan*.

Our calculation also considers that the actual weight of one *dan* increased over time. It increased 3% from the *Han* dynasty to the *Jin* dynasty, doubled from the *Jin* dynasty to the *Sui* dynasty, and then further doubled from the *Tang* dynasty to the *Qing* dynasty [72,73]. The overall increase was about 400% from the *Han* dynasty to the *Qing* dynasty [56]. To be comparable, we convert different units of *dan* to the *Han* convention in our results.

To quantify wage inequality, we use two types of measures: wage ratios (as the baseline measure) and Gini coefficients (as a supplementary measure). The wage ratio between officials and peasants measures the *inter-class* wage inequality, while the wage ratio between high- and low-ranking officials measures the *intra-class* wage inequality within the *Shi* class. Zhang adopts these measures to discuss officials' relative salary in the *Han* dynasty [67]. He estimates that the three councilors' wages were about 47 times higher than those of peasants, whereas state governors' salaries were about 32 times higher. A similar comparison also exists in *Tong Dian*, which documents that in the *Zhou* dynasty the income of an emperor, a high-ranking minister, and a low-ranking official were respectively 320 times, 32 times and 8 times higher than that of a peasant. The wage differential between officials and peasants is the historical counterpart to the public sector wage premium, which is only about 10% in modern times [74].

A partial Gini coefficient can be estimated for the official population and for the official-peasant population. It is supposed to describe the income distribution than wage ratios by considering the weights of population with different wages. However, we must point out that we do not have accurate data on population shares of officials and peasants. Shares within officials are better recorded by historical records (Fig 3) but shares of officials in the entire society were not available. So, our Gini coefficients are at best a *partial* measure of wage inequality for the official-peasant population (the majority of society), rather than for the entire society. Furthermore, the notion of the "average peasant" in our calculation ignores intra-peasant heterogeneities and regional variations pointed out by Pomeranz [75]. It can lead to underestimation of inequality. Thus, the partial Gini coefficient is more a lower bound measure of inequality.

For the proportions of officials at each rank, we follow Bai's estimates (Fig 3) and assume that the relative proportions (not absolute numbers) of officials at different ranks remained stable [76,77]. Nevertheless, there are no direct datasets available for the proportions of the *Shi* class in the population, but we know from historical fiscal records that officials with ranks were about 1% of the total population in all dynasties. Therefore, we use 2% as the *lower bound* of the class to account for the fact that about half of scholars had no ranks such as grassroots officials ("*Li*") and professional academics.

## Results

### Official wage

Fig 4 plots the average wages of officials with ranks and the *Shi* class (full details in Table 2). We cannot distinguish aristocrat officials from *Shi* officials, but the population of aristocrats was ignorable compared to that of the entire *Shi* class. For simplicity, we treat them as one group in calculating the average wage.

The average wage of officials (and the *Shi* class in general) features an inverted U shape over the two millennia. In the first millennium (*Han, Jin, Southern & Northern, Sui, Tang*), average official wage kept rising. In the second millennium (*Song, Yuan, Ming*, and *Qing*), official wages underwent a secular decline. Officials in the *Tang* dynasty earned the highest salaries as

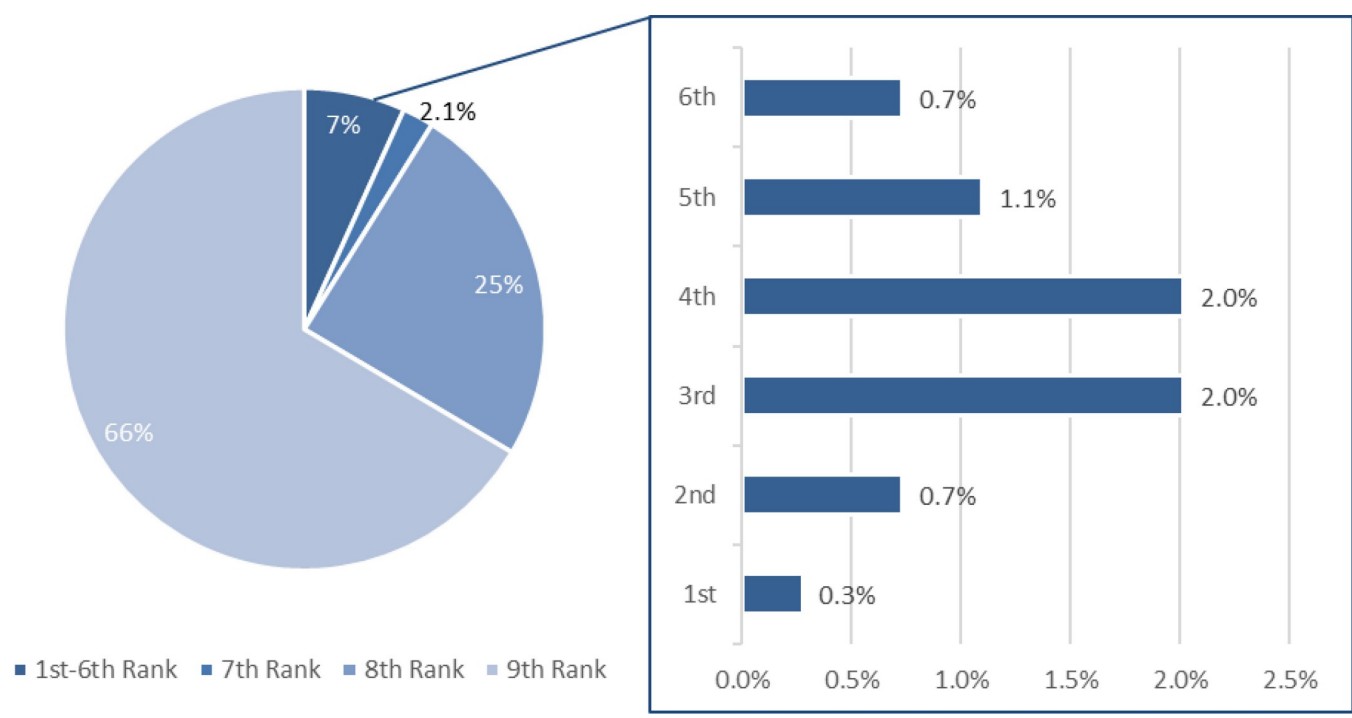

**Fig 3.**

found by previous literature [53,56,57]. Also, the *Tang* dynasty implemented a salary reform to clearly define all hidden forms of income such as food and servant stipends. Similar payroll structure was maintained until the end of the *Song* dynasty. In contrast, officials in later dynasties lost substantial income from farmlands. A significant portion of their income was generated from the rent collected from these farmlands [78]. Paper money was one of the major forms of salary payment in the *Yuan*, *Ming*, and *Qing* dynasties [60]. High inflation significantly reduced the real purchasing power of salaries in the form of paper money.

Along the trends of official wage, two permanent changes occurred. The first was technological progress, which led to an increase in agricultural productivity from the *Han* dynasty to the *Tang* dynasty [79]. This change was responsible for the rising trend of official wage in the first millennium. However, productivity became stagnant in later dynasties until the 19th century when China was forced to open to and learn from the western civilization [39]. The second was the secular concentration of power towards the emperor. From the *Sui* dynasty onwards, the institutionalization of imperial examinations exacerbated intra-class competition and granted the emperor greater bargaining power. The Neo-Confucianism movement from the *Song* dynasty introduced new social norms that further curtailed the political power of officials [40]. These changes in explicit and implicit rules played a significant role in the declining trend of official wage in the second millennium.

## Wage inequality

Following the methods in Section 3.2, the wage ratios are computed and presented in Figs 5 and 6. It is noted that the ratio between official and peasant wages (inter-class wage inequality) resembles the inverted U shape as in the official wage series. We confirm the quadratic relationship using the simple OLS regression, as shown in column (1) of Table 3. As argued earlier, throughout the imperial epoch, peasant wage remained relatively stable at the subsistence

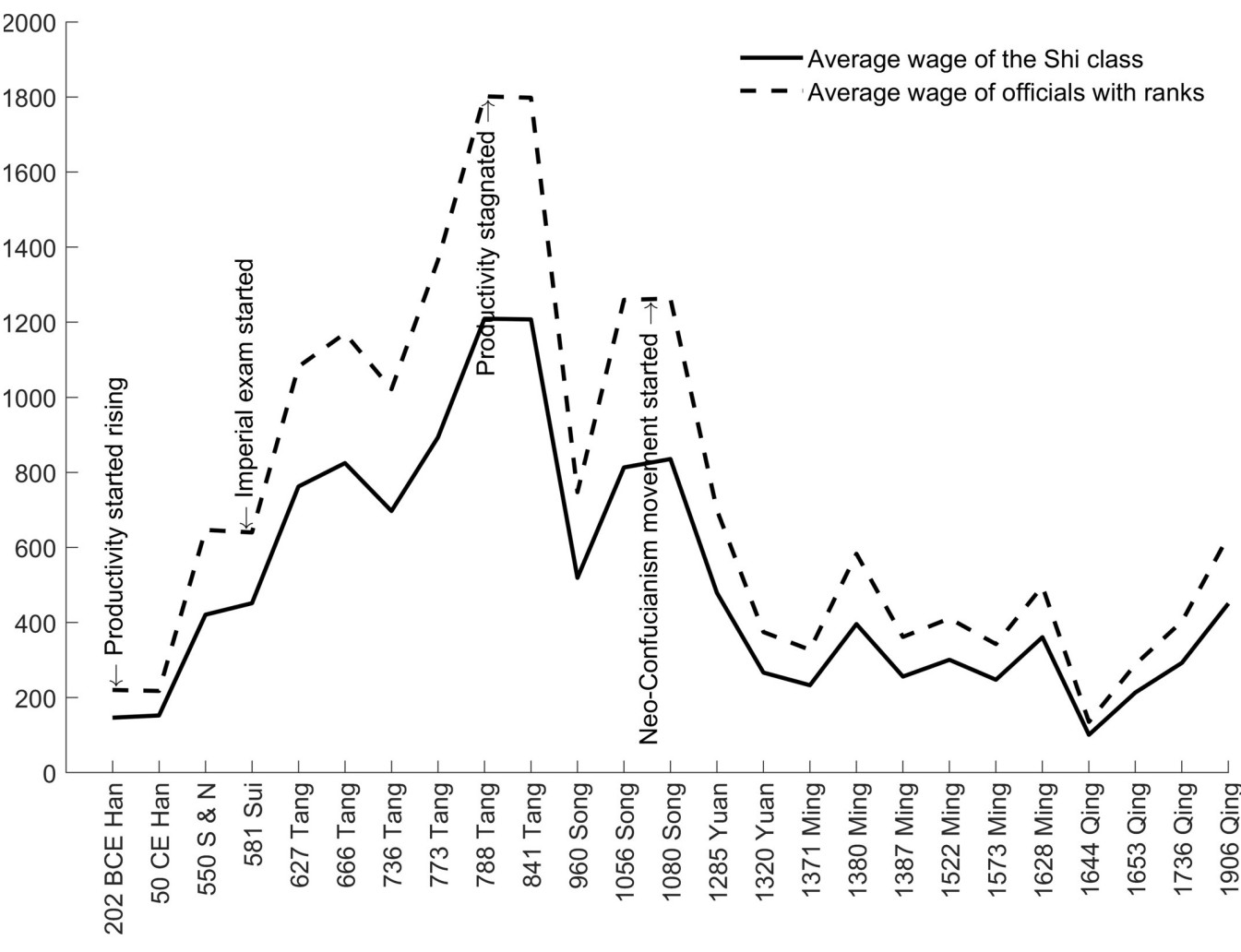

**Fig 4.**

level, showing little increase [26,39,80]. Specifically, recent evidence confirms that the real day wage in the Ming and Qing dynasties remained stable [36].

It is worth emphasizing that peasant wage is *not* the same as peasant output. Benefits of productivity improvements were either reaped by the ruling class of the empire (the emperor, aristocrats, and officials) or cancelled out by population expansion (the "Malthusian checks"). Therefore, the findings of Broadberry et al. on GDP per capita (output) in China do not refute our assumption on wage [81]. The inverted U relationship is demonstrated by the quadratic regression curve in Fig 5. As argued earlier, the initial increase was mainly due to the improvement in agricultural productivity (the technological mechanism), and the later decrease was mainly due to the continued concentration of political power (the political mechanism).

As the evidence for the political mechanism, the ratio between top- and bottom-ranking official wages (intra-class wage inequality) shows a secular declining trend in Fig 6 and column (2) of Table 3. Behind the narrower pay gap within officials were dynamics of emperor-bureaucracy power tensions. Both institutional arrangements and social norms are embodiment of the power tensions. For example, the introduction of imperial exams during the *Sui* dynasty led to increased mobility among officials, resulting in higher competition in the *Shi*

**Table 2. Annual wages of officials and peasants (Unit: *Han* dan).**

| Dynasty | Year | 1st rank | 2nd rank | 3rd rank | 4th rank | 5th rank | 6th rank | 7th rank | 8th rank | 9th rank | *Shi* | Peasant |
|---|---|---|---|---|---|---|---|---|---|---|---|---|
| Han | 202 BCE* | 4320 | 2880 | 1440 | 1152 | 864 | 662 | 432 | 144 | 115 | 73 | 30 |
| Han | 50 CE | 2520 | 1296 | 864 | 720 | 576 | 504 | 504 | 216 | 144 | 87 | 30 |
| Southern & Northern | 550* | 10176 | 7632 | 5088 | 3056 | 2032 | 1266 | 760 | 456 | 360 | 195 | 30 |
| Sui | 581* | 5118 | 4083 | 3048 | 2012 | 1452 | 893 | 714 | 536 | 489 | 263 | 37 |
| Tang | 627* | 5808 | 4710 | 3981 | 3147 | 2526 | 1584 | 1364 | 1040 | 849 | 443 | 37 |
| Tang | 666 | 7320 | 5790 | 4305 | 3147 | 2526 | 1584 | 1364 | 1130 | 921 | 479 | 37 |
| Tang | 736 | 8940 | 6870 | 5205 | 3771 | 2886 | 1674 | 1337 | 909 | 708 | 373 | 37 |
| Tang | 773 | 12000 | 11190 | 6465 | 4875 | 4470 | 2565 | 2048 | 1523 | 807 | 422 | 37 |
| Tang | 788 | 13646 | 10779 | 7288 | 5749 | 5344 | 3338 | 2665 | 1985 | 1197 | 617 | 37 |
| Tang | 841* | 13646 | 10264 | 7288 | 5749 | 5344 | 3338 | 2665 | 1985 | 1197 | 617 | 37 |
| Song | 960* | 5742 | 4356 | 3482 | 2617 | 1954 | 1142 | 936 | 673 | 545 | 291 | 37 |
| Song | 1056 | 20589 | 15983 | 8675 | 3823 | 2716 | 2072 | 1505 | 1200 | 696 | 367 | 37 |
| Song | 1080 | 19529 | 12455 | 9083 | 2993 | 2303 | 1933 | 1419 | 1172 | 779 | 408 | 37 |
| Yuan | 1285 | 4800 | 3600 | 2960 | 2000 | 1440 | 1120 | 960 | 800 | 480 | 259 | 37 |
| Yuan | 1320 | 2400 | 1800 | 1400 | 1000 | 720 | 560 | 480 | 400 | 280 | 159 | 37 |
| Ming | 1371 | 3600 | 2400 | 1600 | 1080 | 720 | 400 | 320 | 280 | 240 | 139 | 37 |
| Ming | 1380 | 5200 | 4400 | 3600 | 2800 | 1600 | 840 | 640 | 480 | 380 | 209 | 37 |
| Ming | 1387 | 4176 | 2928 | 1680 | 1152 | 768 | 480 | 360 | 312 | 264 | 151 | 37 |
| Ming | 1522* | 2828 | 1982 | 1489 | 1103 | 624 | 530 | 376 | 363 | 344 | 191 | 37 |
| Ming | 1573* | 3568 | 2503 | 1436 | 913 | 629 | 535 | 400 | 284 | 268 | 153 | 37 |
| Ming | 1628* | 4238 | 2971 | 1705 | 1118 | 778 | 661 | 495 | 438 | 414 | 225 | 37 |
| Qing | 1644 | 1028 | 848 | 596 | 384 | 260 | 240 | 180 | 140 | 96 | 67 | 37 |
| Qing | 1653 | 1260 | 1085 | 910 | 733 | 560 | 420 | 316 | 280 | 240 | 139 | 37 |
| Qing | 1736 | 1800 | 1550 | 1300 | 1048 | 800 | 600 | 452 | 400 | 328 | 183 | 37 |
| Qing | 1906 | 2880 | 2400 | 2080 | 1680 | 1280 | 960 | 720 | 640 | 505 | 271 | 37 |

Notes: * The exact year is not mentioned in the original source, so the first year of the reign is used. Data sources are listed in Table 1.

class and a weakened collective power against the emperor. The rise of Neo-Confucianism as a social ideology during the *Song* and *Ming* dynasties reinforced the hierarchical and patriarchal structure of the society. This ideology granted the emperor an elevated and almost divine status.

Compared to wage ratios, Gini coefficients further account for the population shares of different classes of wage. Readers should be warned that the Gini coefficient computed in this section is inaccurate due to the lack of data on population distribution. We can only obtain a "partial" Gini coefficient (only containing peasants and officials) as a robustness check. Even though our Gini (0.27) in the *Qing* dynasty is close to the Gini (0.24) estimated based on the social table of the *Qing* dynasty in 1880 [28]. It provides supporting evidence for the reliability of our benchmarks. As shown in Fig 7, the partial Gini coefficient inherits the inverted U trend of wage ratios illustrated in Fig 5. The initial increase suggests that the positive technological effect dominated the negative political effect in the first millennium. The later decrease in the second millennium indicates that the negative political effects were dominant. The technological effect preceded the political effect because it usually takes longer time for political changes as well as associated institutional and social changes. The fitted curve of the official-peasant Gini is shown in Fig 7, which has an inverted U shape trend like Fig 5. The relationship is also confirmed in column (3) of Table 3.

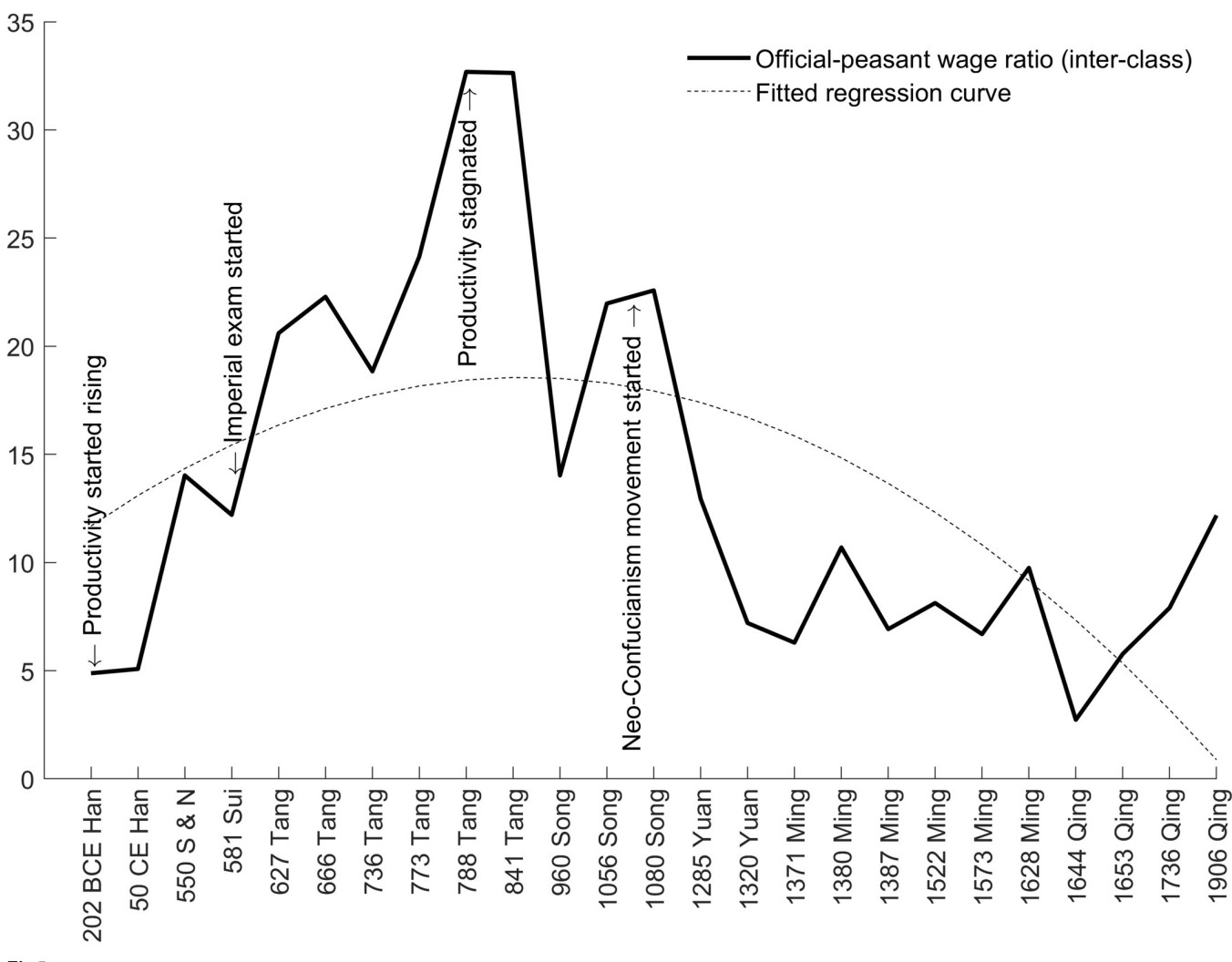

**Fig 5.**

In contrast, the Gini coefficient among high- and low-ranked officials shows a linear declining trend (column (4)) and follows dynastic "inverted u" cycles as in Fig 6. Here we use the lowercase "u" rather than the uppercase "U" to distinguish between short-term cycles from long-term trends. This cyclical pattern is again driven by emperor-bureaucracy power tensions. At the beginning of a new dynasty, the intra-official Gini coefficient was usually low. This is because founding emperors usually implemented a more equal payroll system to consolidate the new empire. One exception was the *Qing* dynasty. This is because the first emperor of the *Qing* dynasty (Shun Zhi) was only 5 years old when he ascended to the imperial position. The dynasty only properly ruled the entire China from the second emperor (Kang Xi) in 1661. Therefore, the actual beginning of the dynasty was much later. In the middle of the dynasties, the intra-official Gini coefficient peaked because wage inequality rose as the power hierarchy expanded and systematic corruption emerged. Towards the end of a dynasty, the intra-official Gini coefficient fell again because uprisings, rebellions, and wars reduced the budget for official salaries. This "inverted u" pattern contributed to the cyclical features. To summarize, the wage inequality features an inverted U trend with inverted u cycles, which we term as the "inverted U-u pattern".

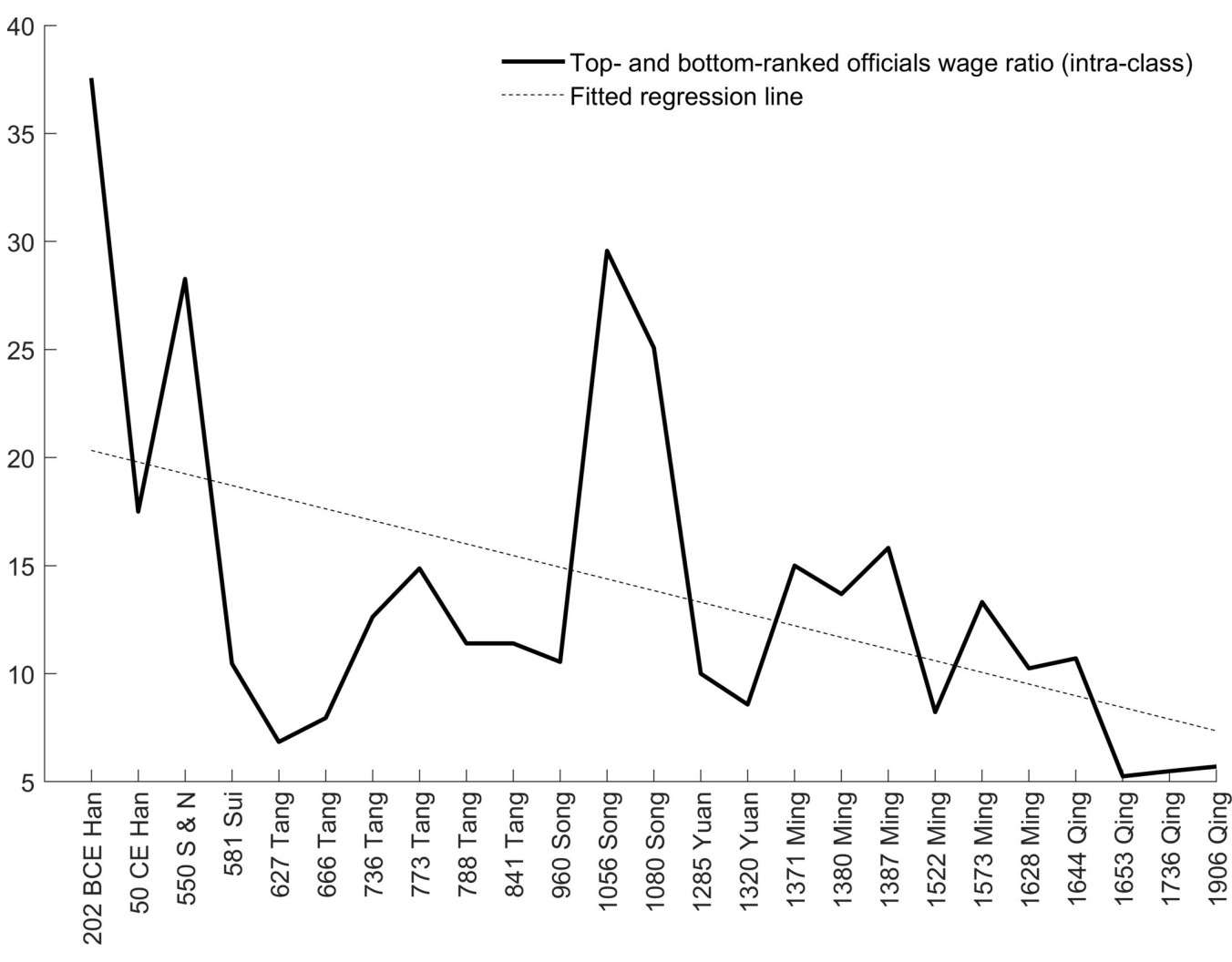

**Fig 6.**

**Table 3. OLS regressions of inter- and intra-class inequality measures.**

|  | (1) | (2) | (3) | (4) |
|---|---|---|---|---|
|  | Official-Peasant Wage Ratio | High-Low Rank Wage Ratio | Official-Peasant Gini | High-Low Rank Gini |
| $t$ | 1.653* | -0.541** | 0.012* | -0.007*** |
|  | (0.827) | (0.201) | (0.006) | (0.002) |
| $t^2$ | -0.081** |  | -0.001** |  |
|  | (0.031) |  | (0.000) |  |
| constant | 10.114** | 20.873*** | 0.304*** | 0.339*** |
|  | (4.668) | (2.991) | (0.033) | (0.028) |

Notes: Standard errors in parentheses. Significance: * 10% ** 5% *** 1%. $t$ is a deterministic time trend.

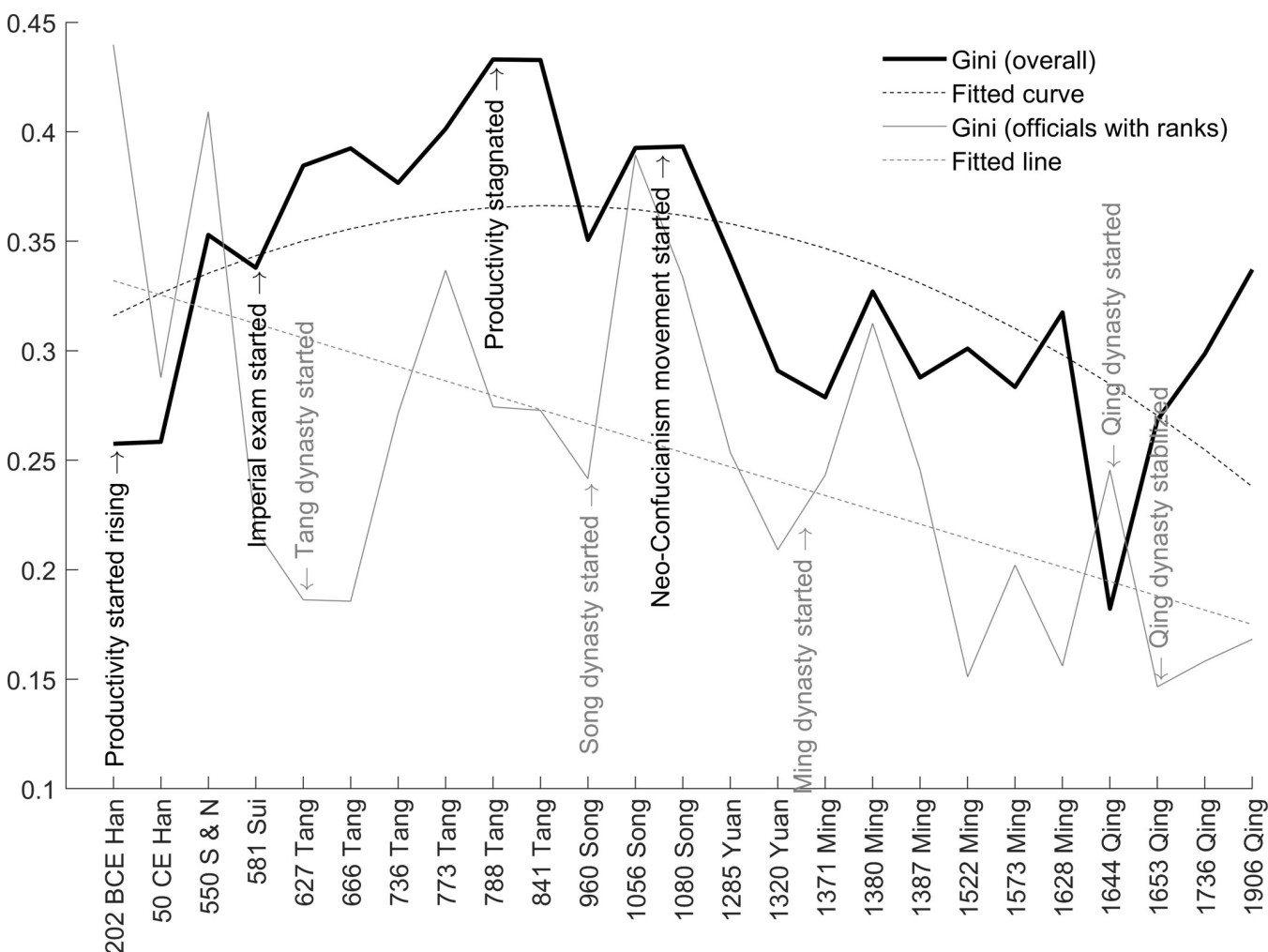

**Fig 7. Partial Gini coefficients of wage.**

## A unified explanation of wage inequality

To understand the inverted U-u pattern, we propose a unified explanation incorporating technological (T), institutional (I), political (P), and social (S) mechanisms—the TIPS framework, inspired by Milanovic (2016). Our framework can be applied to explain both the long-term trend (i.e., the inverted U trend) and short-term cycles (i.e., the inverted u cycles) of wage inequality, as demonstrated in Fig 8.

## The TIPS mechanisms of the "inverted U trends"

(T) The initial push originated from the technological progress during the *Han* dynasty [39,79]. Under the top-down power hierarchy and bottom-up income ladder of imperial China, the gains in productivity were mainly harvested by the upper class. It led to elevated wage inequality in the first millennium, spanning roughly from the *Han* dynasty to the *Tang* dynasty.

(I) To alleviate political tensions arising from the increasing wage inequality, imperial exams were introduced during the *Sui* dynasty [82]. This institutional shift essentially dealt with high wage inequality by promoting social mobility [49]. This reform bolstered the top-

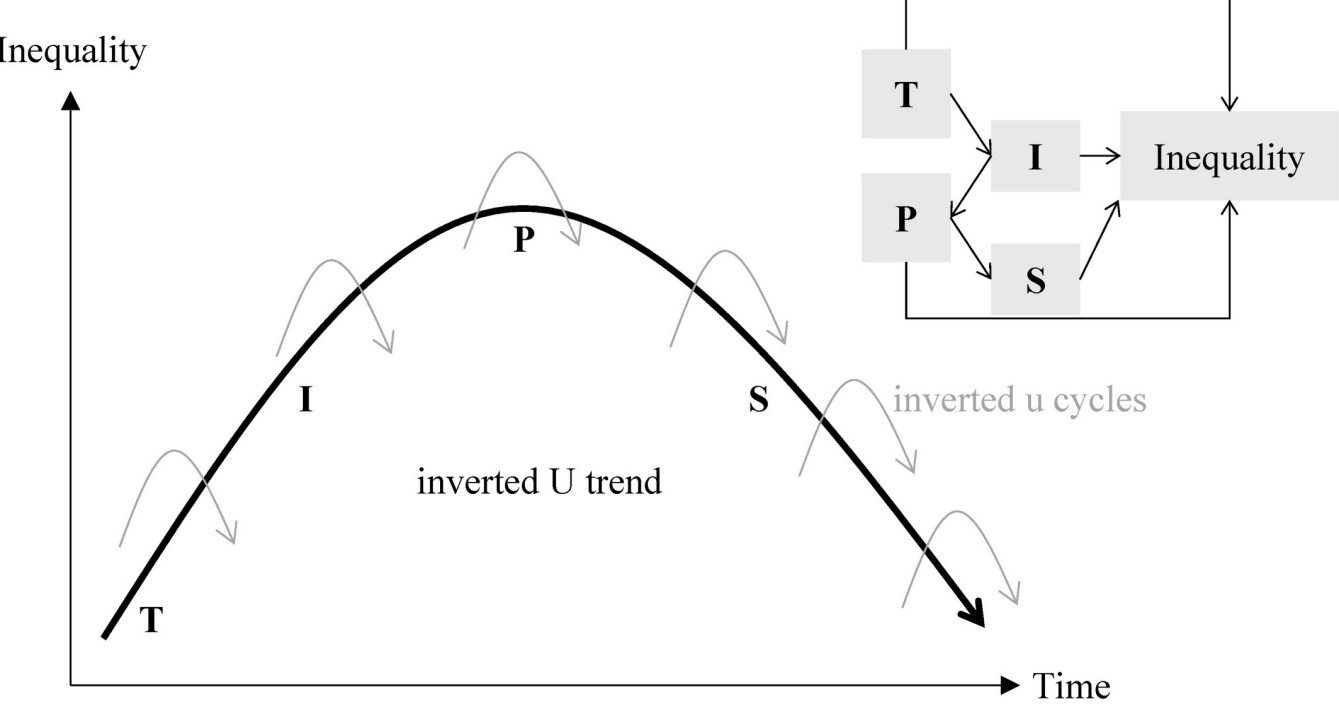

**Fig 8.**

down power hierarchy, as officials (and the overall *Shi* class) were granted greater opportunities to ascend along the income ladder. The consequence was an expansion in the size and authority of bureaucratic officials [43].

(P) The expansion of officials of the empire not only led to political tensions between the emperor and officials, but also between imperial China (especially the *Tang* dynasty) and other empires. On the one hand, conflicts between the emperor and influential officials led to more frequent internal rebellions (e.g., An Lushan Rebellion in 755). On the other hand, external wars became more common as the empire reached its peak in the *Tang* dynasty (e.g., the Eastern Turkic-Tang war in 626, the Baekje-Goguryeo-Silla war in 666, and the Arab-Tang battle of Talas in 751). The surge in military expenses coincided with productivity stagnation, resulting in a decline of wage inequality from late-*Tang* [83].

Besides, political tensions also affected wage inequality in peaceful times. The periods of rising wage inequality often coincided with a politically powerful bureaucracy (e.g., *Han*, *Jin*, *Southern & Northern*). Since imperial exams were introduced in the *Sui* dynasty, the supply of bureaucrats became streamlined, so officials and backup scholars were no longer scarce. The negotiation power of the entire *Shi* class relative to emperors was weakened, contributing to the secular decline in wage inequality (e.g., *Song*, *Ming*, and *Qing*).

(S) Eventually, to address the emperor-bureaucracy tension, social norms rooted in Neo-Confucianism were introduced during the *Song* dynasty [40,84]. It emphasized moral integrity, social hierarchy, and the importance of maintaining social harmony. These values reinforced the idea of meritocratic governance through the "*Ke Ju*" examination system, which promoted the rise of individuals based on knowledge and virtue rather than inherited wealth or status. Improved social mobility then induced competition among officials, which further strengthened the emperor's authority over officials in the *Ming* and *Qing* dynasties. As a result, official

wage and wage inequality underwent a secular decline as the bargaining power of the *Shi* class diminished.

## The TIPS mechanisms of the "inverted u cycles"

The TIPS framework can also be applied to explain inverted u cycles in each dynasty.

(T) New dynasties were usually born out of bottom-up uprisings (e.g., *Han*, *Ming*), internal rebellions (e.g., *Sui*, *Tang*, *Song*), or external wars (e.g., *Yuan*, *Qing*). Therefore, they started with relatively low population levels and high marginal labor productivity due to the law of diminishing marginal product. The positive productivity shocks temporarily increased output per capita at the beginning of each dynasty. The surplus output was either channeled by peasants to support higher fertility rates (the "Malthusian checks") or acquired by the upper class to raise wage inequality.

(I) As the empire developed, its institutions expanded, resulting in a greater number of low-ranking and grassroots officials within the bureaucracy. This increasing scale and complexity often gave rise to inefficiencies, favoritism, and instances of bribery, as officials grappled with navigating intricate administrative procedures and securing their positions. As time passed, specific corrupt practices could take root within the system, as corrupt officials imparted their methods to their successors. The normalization of corruption resulted in a heightened wage inequality, particularly among the officials themselves.

(P) Wage inequality among officials typically began at a low level to solidify the foundations of the new empire. However, in the middle of each dynasty, the central authority of the empire might weaken due to territorial expansion (e.g., *Yuan*) or the proliferation of regional powers (e.g., *Han*). The weakening central power can create opportunities for local officials to engage in corrupt practices, resulting in a reduction of intra-class wage inequality but an elevation of inter-class wage inequality. At the same time, officials could leverage their political influence to negotiate their own benefits with emperors. Numerous historical records bear witness to this. For instance, during the early *Tang* dynasty (634), the imperial secretariat implored the emperor to grant rice salaries to local officials [85]. According to the *Old Book of Tang*, two grand councilors petitioned for a salary increase due to a spike in rice prices in 777 [86]. Similarly, in 787 an official petitioned for a salary raise for all central government officials, highlighting that their salaries significantly lagged those of local officials [85]. Sometimes, emperors also took the initiative to offer salary increases to gain political support from officials. Such adjustments in salaries were manifestations of the power dynamics between emperors and officials. Eventually, the dynasty was either overturned by bottom-up uprisings (e.g., *Ming*), internal rebellions (e.g., *Tang*), or external invasions (e.g., *Song*). Military expenditures in later phases of dynasties decreased output and wage for all classes, resulting in a decline in wage inequality.

(S) As a dynasty lasted over generations, it gave rise to social norms rooted in the legacies of successive rulers. These implicit rules were utilized by the upper class to govern the lower class (e.g., Confucianism's dominance during the *Han* dynasty) or by emperors to constrain officials (e.g., Neo-Confucianism's emergence during the *Song* dynasty). Classical Confucianism before the *Song* dynasty primarily focused on personal and social ethics along with an ideological structure for top-down hierarchical governance. As the "Chinese Renaissance", Neo-Confucianism entrenched the dominant role of Confucianism from the *Song* to the *Qing* dynasties. Essentially, the two schools of thought share the same ethical principles such as loyalty and benevolence in the emperor-official, husband-wife, parent-child, and other interpersonal relationships. The new aspects of Neo-Confucianism were the metaphysical and cosmological concepts such as "Qi" (vital energy) to rationalize those ethical principles. These social norms

contributed to mitigating the political tensions facing the emperor at the cost of the lower class and the "middle class" (officials). It is worth reiterating that the wage inequality shown in Fig 6 does not include emperors, so a lower Gini coefficient could coexist with higher inequality between the emperor and the remaining population.

## Conclusion

This paper attempts to provide some benchmarks of wage inequality in imperial China over two millennia. Based on historical records of salaries and prices, we convert various forms of wage to equivalent rice volumes for officials and peasants. We discover an "inverted U-u" pattern of the wage ratio between officials and peasants, which gradually increased in the first millennium and declined in the second (inverted U trends) with dynastic cycles (inverted u cycles). In contrast, the wage ratio within officials has a secular decline trend.

We propose a unified TIPS theoretical framework to explain these patterns. Changes in both trends and cycles started with technological disruptions (T) which led to higher wage inequality. Then, institutional arrangements (I) adapted to address the consequences of higher inequality but further exacerbated it. As a result, political tensions (P) grew between the upper and lower classes, between the emperor and officials, as well as between imperial China and other empires. Consequently, bottom-up uprisings, internal rebellions, and external wars squeezed the budget for official salaries and reduced wage inequality. Gradually, social norms (S) evolved to reinforce the existing power structure, leading to lower inequality paired with lower mobility, until the empire collapsed from inside or was conquered from outside. The TIPS framework holds for both long-term inverted U trends over the two millennia and short-term inverted u cycles within dynasties.

These factors are still relevant today. For example, the rapid development and deployment of automation and AI in the past few decades (T) have transformed industries such as manufacturing, retail, and services. While these technologies have increased productivity and efficiency, they have disproportionately benefited highly skilled workers in tech and innovation sectors, while displacing lower-skilled workers in more traditional jobs. This has led to a widening wage gap between highly paid professionals and lower-wage workers, increasing overall wage inequality. In response to the growing inequality caused by technological disruption, governments and institutions (I) have implemented various policies, such as tax incentives for tech companies, deregulation of certain industries, and targeted education and reskilling programs. However, these measures have struggled to keep pace with the speed of technological change, leaving many workers behind. As inequality has risen, political tensions (P) have emerged between different socio-economic groups. In many countries, the working class has expressed frustration over stagnant wages and job losses, while elites in the tech and finance sectors have thrived. These tensions have manifested in populist movements, protests, and rising polarization in political discourse. In some cases, these movements have challenged traditional political elites, leading to the election of outsider candidates and shifts in policy priorities. Over time, social norms (S) have adapted to these changes, reinforcing existing power structures. In some societies, the narrative of technological progress has been used to justify growing inequality, with arguments that innovation and efficiency will eventually "trickle down" to benefit all. This has led to greater acceptance of income and wealth disparities, and in some cases, has slowed efforts to reduce inequality through stronger redistributive policies. Nevertheless, this also results in lower social mobility, as access to opportunities becomes increasingly tied to one's position in the socio-economic hierarchy.

Therefore, current policymakers are suggested to be proactive and keep updated in their interventions to counteract inequality arising from new technologies. Progressive tax policies,

redistribution mechanisms, and investments in education and reskilling programs could help ensure that the benefits of technological innovation are more evenly distributed.

One limitation of our method is that we place a higher priority on the validity and comparability of data sources. This approach comes at the expense of excluding numerous unofficial data sources, such as genealogies [48], folk stories, and poems [53], which could provide broader coverage of social classes and sample periods. Nevertheless, our estimates can serve as benchmarks for future studies when new sources of data become available. In addition, we use the wage ratio between officials and peasants to measure inter-class wage inequality. However, it is important to consider that as commerce and industry developed, particularly from the Tang dynasty onwards, the economic structure became more diversified, and the relevance of peasant income as a baseline for measuring overall wage inequality may have shifted. The growth of urban centers and the expansion of trade and industrial activities created new sources of wealth and income beyond agriculture. As a result, merchants, artisans, and industrial workers increasingly contributed to the economy, potentially altering the dynamics of income distribution. A more comprehensive measure of wage inequality would need to consider the emerging wage disparities within commerce and industry, as well as the interactions between these sectors and traditional agriculture.

## Supporting information

**S1 File. Data and code can be found in the zip file of the supporting information.**
(ZIP)

## Author Contributions

**Conceptualization:** Qiang Wu, Guangyu Tong, Peng Zhou.

**Data curation:** Qiang Wu, Guangyu Tong, Peng Zhou.

**Formal analysis:** Qiang Wu, Guangyu Tong, Peng Zhou.

**Funding acquisition:** Qiang Wu.

**Investigation:** Qiang Wu, Guangyu Tong, Peng Zhou.

**Methodology:** Qiang Wu, Peng Zhou.

**Project administration:** Qiang Wu, Peng Zhou.

**Resources:** Qiang Wu, Guangyu Tong.

**Software:** Qiang Wu, Peng Zhou.

**Supervision:** Peng Zhou.

**Validation:** Qiang Wu, Guangyu Tong, Peng Zhou.

**Visualization:** Peng Zhou.

**Writing – original draft:** Qiang Wu, Guangyu Tong, Peng Zhou.

**Writing – review & editing:** Peng Zhou.

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
