## [Decision Letter · Decision Letter 0]

17 Oct 2024

PONE-D-24-15693Long-term Wage Inequality in Imperial China: From 202 BCE to 1912 CEPLOS ONE

Dear Dr. Zhou,

Thank you for submitting your manuscript to PLOS ONE. After careful consideration, we feel that it has merit but does not fully meet PLOS ONE’s publication criteria as it currently stands. Therefore, we invite you to submit a revised version of the manuscript that addresses the points raised during the review process.

We look forward to receiving your revised manuscript.

Kind regards,

Shai Gordin, Ph.D.

Academic Editor

PLOS ONE

Journal Requirements:

 This study was funded by the Ministry of Education in China Project of Humanities and So-cial Sciences (Project No. 19YJA790089).  

3. Please ensure that you refer to Figure 7 in your text as, if accepted, production will need this reference to link the reader to the figure.

Reviewers' comments:

Reviewer's Responses to Questions

**Comments to the Author**

1. Is the manuscript technically sound, and do the data support the conclusions?

Reviewer #1: Yes

Reviewer #2: Yes

2. Has the statistical analysis been performed appropriately and rigorously? 

Reviewer #1: Yes

Reviewer #2: Yes

3. Have the authors made all data underlying the findings in their manuscript fully available?

Reviewer #1: Yes

Reviewer #2: Yes

4. Is the manuscript presented in an intelligible fashion and written in standard English?

Reviewer #1: Yes

Reviewer #2: Yes

5. Review Comments to the Author

Reviewer #1: This study compiles historical records to benchmark wage inequality in imperial China from 202 BCE to 1912 CE. The authors find a gradual increase in wage inequality from the Han to the mid-Tang dynasty and a decline from the late-Tang to the Qing dynasty, presenting an overall inverted U-shaped pattern, nested with short-term “inverted u” cycles. Contrarily, wage inequality among officials exhibited a secular decline over two millennia. The paper also proposes a unified theoretical framework to explain inequality evolution in pre-industrial society.

The topic of this paper is intriguing. The extensive historical span and comprehensive data analysis provide valuable insights into the long-term wage inequality in imperial China. I recommend the editor to consider a quick minor revision for publication. Here are three minor comments to enhance the manuscript.

Comment 1: The discussion on social mobility would benefit from distinguishing between intergenerational and intragenerational mobility. For example, the “Ke Ju” system might have promoted intergenerational mobility by allowing young individuals from lower-class families to ascend social ranks through the examination system, thereby reducing inequality. This aligns with the observed trend of decreasing inequality after the peak during the Sui-Tang period. Conversely, the evolution of social norms beginning in the Song dynasty may have restricted rights, potentially reducing intragenerational mobility. This observation corresponds with Yang & Zhou’s (2022) findings on the positive correlation between inequality and intragenerational mobility (Gatesby curve).

Comment 2: On page 10, the manuscript states, “We use equivalent volumes of rice to measure the real wage.” However, rice yields would have increased with agricultural advancements, potentially affecting the measurement of both peasants’ income and officials’ land rent income. The authors should provide additional clarification, such as explaining that agricultural output is distributed between officials and farmers through land rent, and therefore, the increase in unit land yield affects both groups proportionally. Furthermore, discussing the impact of rising land rent (relative share r/(1-r)) on the distribution of agricultural output between the two classes could be beneficial.

Comment 3: The paper measures inter-class wage inequality using “the wage ratio between officials and peasants”. As commerce and industry developed since the Tang dynasty, how might wage inequality measured the relative wage indicator based on peasant income be influenced? A brief discussion on this point would be helpful.

Reviewer #2: Peer Review:

Long-term Wage Inequality in Imperial China: From 202 BCE to 1912 CE

Summary of the Paper

This paper examines the evolution of wage inequality in Imperial China over a span of more than two millennia, from the Han dynasty (202 BCE) to the Qing dynasty (1912 CE). The study uses historical records to compare wage levels between different social classes, focusing primarily on officials and peasants. By converting salaries into comparable units of rice, the study creates a long-term view of wage inequality, which is then analysed using salary ratios and partial Gini coefficients. The paper introduces a TIPS framework—Technological, Institutional, Political, Social mechanisms—to explain the fluctuations in wage inequality over this period. The findings show that wage inequality between officials and peasants followed a cyclical pattern, with an overall "inverted U" shape over the 2,000 years. Technological advancements increased inequality, while institutional reforms and social norms reduced it.

Comments

1. Extensive Historical Data and Unique Long-term Perspective: The paper provides a rare, comprehensive analysis of wage inequality over an exceptionally long period, drawing on more than two millennia of historical data. This long-term perspective is a major strength of the research, as it allows the authors to identify long-term patterns and trends that would be impossible to discern from shorter timeframes. This makes the paper a valuable contribution to both economic history and inequality studies.

2. Innovative Theoretical Framework: The TIPS framework is an innovative tool for understanding the drivers of wage inequality over such a long historical period. By integrating technological, institutional, political, and social factors, the paper offers a multifaceted explanation for both long-term trends and short-term cycles in inequality. This framework could potentially be applied to other regions or periods, making it a model for future research in economic history.

3. Solid Methodological Approach: The method of converting salaries from various forms into rice equivalents provides a standardised way to measure wages across different centuries and dynasties. This adds a layer of methodological rigor that strengthens the validity of the comparisons and trends identified in the paper. Additionally, the use of salary ratios and Gini coefficients offers a reliable quantitative approach to measuring inequality.

Suggestions for Improvement

1. Clarify Definitions Early On: The paper discusses terms like "wage inequality" and "intra-class" vs. "inter-class" inequality but could benefit from clearer definitions of these terms early in the paper. A brief explanation of how these concepts are measured (e.g., salary ratios, Gini coefficients) right in the introduction would improve clarity for readers unfamiliar with the technical jargon.

2. Clarify the Role of Social Norms: The paper mentions Neo-Confucianism and other social norms as factors that reduced inequality, but this discussion could be further elaborated. Specifically, the mechanisms through which these social norms influenced wage structures could be clarified and backed by additional historical sources or examples. This would help to strengthen the theoretical argument and provide clearer connections between social norms and economic outcomes.

3. Include Practical Implications: While the historical context is well-explained, the paper could briefly mention what the findings mean for modern-day inequality studies. A short section or paragraph discussing how this historical analysis could inform current policymaking or research on inequality would provide more relevance to contemporary readers.

4. Other minor comments: missing information or typo, such as Page 3 first paragraph, an empty parenthesis. Add short explanations under tables and figures for better readability

6. PLOS authors have the option to publish the peer review history of their article (what does this mean?). If published, this will include your full peer review and any attached files.

Reviewer #1: No

Reviewer #2: **Yes: **Chaowei Wang

---

## [Author Response · Author response to Decision Letter 0]

21 Oct 2024

Please see our responses in the attached document: response to reviewers.

---

## [Decision Letter · Decision Letter 1]

28 Nov 2024

Long-term Wage Inequality in Imperial China: From 202 BCE to 1912 CE

PONE-D-24-15693R1

Dear Dr. Zhou,

We’re pleased to inform you that your manuscript has been judged scientifically suitable for publication and will be formally accepted for publication once it meets all outstanding technical requirements.

Kind regards,

Shai Gordin, Ph.D.

Academic Editor

PLOS ONE

Additional Editor Comments (optional):

Reviewers' comments:

Reviewer's Responses to Questions

**Comments to the Author**

1. If the authors have adequately addressed your comments raised in a previous round of review and you feel that this manuscript is now acceptable for publication, you may indicate that here to bypass the “Comments to the Author” section, enter your conflict of interest statement in the “Confidential to Editor” section, and submit your "Accept" recommendation.

Reviewer #1: All comments have been addressed

Reviewer #2: All comments have been addressed

2. Is the manuscript technically sound, and do the data support the conclusions?

Reviewer #1: Yes

Reviewer #2: Yes

3. Has the statistical analysis been performed appropriately and rigorously? 

Reviewer #1: Yes

Reviewer #2: Yes

4. Have the authors made all data underlying the findings in their manuscript fully available?

Reviewer #1: Yes

Reviewer #2: Yes

5. Is the manuscript presented in an intelligible fashion and written in standard English?

Reviewer #1: Yes

Reviewer #2: Yes

6. Review Comments to the Author

Reviewer #1: I am pleased to see that the authors have thoroughly addressed all of my previous comments in the revised manuscript. I now believe the paper is ready for publication and recommend accepting it. Congratulations to the authors on completing an excellent piece of research!

Reviewer #2: All comments and suggestions from the previous reviews have been carefully considered and implemented.

7. PLOS authors have the option to publish the peer review history of their article (what does this mean?). If published, this will include your full peer review and any attached files.

Reviewer #1: No

Reviewer #2: No

---

## [Editor Report · Acceptance letter]

13 Dec 2024

PONE-D-24-15693R1 

PLOS ONE

Dear Dr. Zhou, 

I'm pleased to inform you that your manuscript has been deemed suitable for publication in PLOS ONE. Congratulations! Your manuscript is now being handed over to our production team.

Kind regards, 

on behalf of

Dr. Shai Gordin 

Academic Editor

PLOS ONE